# Altered ubiquitin signaling induces Alzheimer's disease-like hallmarks in a three-dimensional human neural cell culture model

Inbal Maniv[1,7], Mahasen Sarji[1,7], Anwar Bdarneh[1,7], Alona Feldman[1], Roi Ankawa[1], Elle Koren[1], Inbar Magid-Gold[1], Noa Reis[1], Despina Soteriou[1], Shiran Salomon-Zimri[2], Tali Lavy[1], Ellina Kesselman[3], Naama Koifman[3], Thimo Kurz [4], Oded Kleifeld [1], Daniel Michaelson[2], Fred W. van Leeuwen[5,8], Bert M. Verheijen[1,5], Yaron Fuchs [1,6] ✉ & Michael H. Glickman [1] ✉

Alzheimer's disease (AD) is characterized by toxic protein accumulation in the brain. Ubiquitination is essential for protein clearance in cells, making altered ubiquitin signaling crucial in AD development. A defective variant, ubiquitin B + 1 (UBB[+1]), created by a non-hereditary RNA frameshift mutation, is found in all AD patient brains post-mortem. We now detect UBB[+1] in human brains during early AD stages. Our study employs a 3D neural culture platform derived from human neural progenitors, demonstrating that UBB[+1] alone induces extracellular amyloid-β (Aβ) deposits and insoluble hyperphosphorylated tau aggregates. UBB[+1] competes with ubiquitin for binding to the deubiquitinating enzyme UCHL1, leading to elevated levels of amyloid precursor protein (APP), secreted Aβ peptides, and Aβ build-up. Crucially, silencing UBB[+1] expression impedes the emergence of AD hallmarks in this model system. Our findings highlight the significance of ubiquitin signalling as a variable contributing to AD pathology and present a nonclinical platform for testing potential therapeutics.

Alzheimer's disease (AD) is a devastating neurodegenerative disease that is the main cause of dementia. Currently, no effective treatments that modify disease progression are available (although anti-amyloid antibodies have shown potential clinical benefits in early AD patients[1]). AD is defined neuropathologically by two major lesions: (i) extracellular plaques containing amyloid-β (Aβ) and (ii) intracellular tau aggregates known as neurofibrillary tangles (NFTs)[2–4]. Although rare familial AD (FAD) cases are caused by inherited mutations in specific genes, the proximal cause of sporadic AD (SAD) remains unknown. Insufficient clearance of neurotoxic proteins (i.e., Aβ and tau) has been suggested to underlie the majority of SAD cases[2,3]. Therefore, exploring the roles of protein clearance pathways in AD may bring new insights into mechanisms of AD pathogenesis and could provide novel therapeutic avenues.

[1]Department of Biology, Technion Israel Institute of Technology, Haifa 3200003, Israel. [2]Department of Neurobiology, The George S. Wise Faculty of Life Sciences, The Sagol School of Neuroscience, Tel Aviv University, Tel Aviv 69978, Israel. [3]The Wolfson Department of Chemical Engineering, The Technion Center for Electron Microscopy of Soft Matter, Technion Israel Institute of Technology, Haifa 3200003, Israel. [4]School of Molecular Biosciences, University of Glasgow, Glasgow G12 8QQ Scotland, UK. [5]Department of Neuroscience, Maastricht University, 6229 ER Maastricht, the Netherlands. [6]Present address: Augmanity, Rehovot 7670308, Israel. [7]These authors contributed equally: Inbal Maniv, Mahasen Sarji, Anwar Bdarneh. [8]Deceased: Fred W. van Leeuwen. ✉e-mail: yaronfox@gmail.com; glickman@technion.ac.il

Most protein clearance from cells is ubiquitin-dependent. Ubiquitin is a small (76 amino acids) protein that determines the fate and integrity of proteins within the cell and is implicated in a variety of human diseases, including AD[5-8]. For instance, ubiquitin's ability to target proteins for degradation by the proteasome, a multi-subunit protease complex, is critical for neuronal proteostasis[9]. There are no known naturally occurring mutations in ubiquitin-coding genes, yet a ubiquitin variant can arise from a rare transcriptional error, termed ubiquitin B+1 (UBB+1)[10]. This unique non-inherited frameshift results in an amino acid change at position 76 of ubiquitin replacing glycine at the C-terminus followed by a missense 19 amino acid extension that abolishes its ability to modify proteins[10-13]. For this reason, UBB+1 cannot serve as a molecular signal to alter the fate of proteins that are typically targeted for ubiquitination. Since cells have difficulty removing UBB+1, its levels can build up over time potentially interfering with ubiquitin signaling by binding to components of the ubiquitin system[10-13]. Intriguingly, UBB+1 has been found in 100% of AD patients' brains at post-mortem examination[10]. However, a causal connection between UBB+1 and AD has never been established, partly due to the lack of adequate AD models[14,15]. As AD may be a uniquely human condition[16], the study of AD pathomechanisms may require models that accurately mimic the disease processes seen in patients[15,17-19].

In the present study, we identify UBB+1 in human brains and several model systems even at the earliest stages of AD pathogenesis. We use a 3D human neural cell culture model[20] to explore the effects of UBB+1 expression in a disease-relevant context. We show that expression of UBB+1 is sufficient to trigger AD-like pathology in this model, giving rise to plaques containing Aβ and phosphorylated tau aggregates reminiscent of NFTs. Furthermore, we demonstrate that UBB+1 competes with ubiquitin for binding to ubiquitin C-terminal hydrolase-L1 (UCHL1), a deubiquitinating enzyme (DUB) that is highly abundant in neurons and is important for the turnover of the amyloid precursor protein (APP). Importantly, we find that silencing UBB+1 expression in a 3D human cell culture expressing FAD mutations is sufficient to suppress the emergence of AD-like pathology.

## Results

### Accumulation of UBB+1 is an early event in Alzheimer's disease

The correlation between UBB+1 and AD hallmarks is well-established. Nevertheless, it remains unclear whether the presence of the UBB+1 protein itself is an underlying cause or a result of neurodegeneration. To gain insight into the nature of this relationship, we searched for evidence of UBB+1 at the early stages of AD. Using human post-mortem brain tissue from donors diagnosed at different Braak stages, we stained the entorhinal cortex using a UBB+1-specific antibody. The entorhinal cortex is among the earliest brain regions to deteriorate in AD and is therefore suitable for exploring early pathological changes[21]. Aggregate-shaped structures immunoreactive for UBB+1 were frequent in AD brains at Braak V-VI and were visualized by immunofluorescence (IF) even at early Braak stages (Braak I-II) (Fig. 1a). Quantification of UBB+1-positive aggregates showed a significant increase along Braak stages starting at Braak II (Fig.1b). Similar results were obtained by immunohistochemistry (IHC) of samples from the same tissues (Fig.1c).

Having found that the accumulation of UBB+1 is an early event in AD, we set out to investigate whether UBB+1 also accumulates in different AD models. Apolipoprotein E (APOE) is a cholesterol/lipid transporter that has been strongly linked to AD pathogenesis[22-24]. APOE4, in contrast to the "neutral" APOE3 isoform, is the strongest genetic risk factor for sporadic AD and was previously shown to increase Aβ production and tau phosphorylation in human neurons[25]. This isoform was used to exacerbate neurodegeneration in transgenic humanized AD mice[26]. To explore whether UBB+1 is a feature associated with APOE4, we evaluated the presence of UBB+1 in the ApoE4-targeted replacement mouse model of AD (ApoE4-TR)[27]. First, we obtained lysates from the hippocampus of 4-month-old ApoE3/4-TR mice and immunoblotted for UBB+1 (Fig. 1d). ApoE4 mice samples contained significantly increased levels of the UBB+1 protein as compared to ApoE3 control mice (Fig. 1e). To gain further insight into this finding on increased levels of UBB+1 in ApoE4-TR mice, we examined the presence of UBB+1 in hippocampal sections by immunofluorescence staining and compared it to ApoE3-TR controls. We identified increased UBB+1-positive staining mostly in the dentate gyrus (DG) area of the hippocampus of 1-month-old ApoE4-TR mice (Fig. 1f), an early timepoint in the progression of AD-like pathogenesis[27,28]. UBB+1 expression was found to increase with age and was prominently present by 4 months (Fig. 1g). Co-staining with the neuronal marker microtubule-associated protein 2 (MAP2), and the astrocyte marker glial fibrillary acidic protein (GFAP), revealed that UBB+1 is specifically localized to neurons but not astrocytes (Fig. 1f,g). While ApoE4 correlates with elevated UBB+1 at early stages of AD-like pathogenesis in this mouse model, it remains unclear how the ApoE genotype contributes to UBB+1 accumulation.

Next, we investigated whether UBB+1 accumulates in human cells that carry FAD mutations. SK-N-SH neuroblastoma cells were transduced with vectors expressing mutations in genes that are found in FAD: human APP with both K670N/M671L (Swedish) and V717I (London) mutations (APP^SL) as well as presenilin 1 with the ΔE9 mutation (PSEN1ΔE9). Accumulated UBB+1 was detected in cytoplasmic puncta, with ~6-fold more puncta in APP^SL/PSEN1^ΔE9 cells as compared to control cells (Supplementary Fig. 1a,b). Immunoblot analysis of SK-N-SH lysates verified the presence of the UBB+1 protein in APP^SL/PSEN1^ΔE9 cells (Supplementary Fig. 1c). UBB+1 puncta also accumulated in APP^SL/ PSEN1^ΔE9 SK-N-SH cells upon differentiation (Supplementary Fig. 1d, e).

An elegant 3D human neural cell culture system carrying FAD mutations (APP^SL/PSEN1^ΔE9) has been developed as a platform that enables the recapitulation of both Aβ plaques and NFT pathology[19,20,29]. We followed this protocol to culture human neuronal cells in a 3D Matrigel matrix and examined them for the presence of the UBB+1 protein. By 6 weeks, UBB+1-positive staining was significantly more abundant in these FAD 3D cultures (APP^SL/PSEN1^ΔE9/mCherry) than in control cultures (Supplementary Fig. 1f,g). While FAD mutations correlate with elevated UBB+1 in a variety of human tissue cultures, it remains unclear how APP or PSEN1 genotypes contribute to UBB+1 accumulation.

Together, these results demonstrate that UBB+1 accumulation occurs early on in AD pathogenesis, and is an early and common feature in several well-studied AD models.

### UBB+1 expression induces AD pathology in human neurons

The above-mentioned 3D human neuronal platform provided an opportunity to study the roles of UBB+1 and altered ubiquitin signaling in the development and progression of AD. In order to examine the role of UBB+1 in AD pathogenesis we set up a similar system by transducing neural progenitor ReN cells with an mCherry lentiviral vector encoding UBB+1 and performing cell sorting for mCherry+ cells (Fig. 2a). Cells with the top 2% fluorescence signal intensity (indicating efficient expression) were collected and used in the following experiments (Supplementary Fig. 2a). In parallel, we established a separate cell line similar to the published FAD model[19,20,29] by transducing ReN cells with mCherry lentiviral vectors encoding for human APP^SL coupled with the expression of PSEN1^ΔE9 (APP^SL/PSEN1^ΔE9, referred to herein as FAD). Confocal microscopy indicated high transduction efficiency of both FAD and UBB+1 expression vectors in their respective lines (Supplementary Fig. 2b). These progenitor cell lines were then analyzed for APP and UBB+1 protein expression by immunoblotting to confirm adequate expression of the encoded proteins (Supplementary Fig. 2c). After embedding in Matrigel and differentiation into 3D neuronal cultures for a duration of 3–12 weeks, various markers of neuronal and glial cells such as NMDA receptor 2B (NR2B), tyrosine hydroxylase (TH), and GFAP indicated proper differentiation (Supplementary

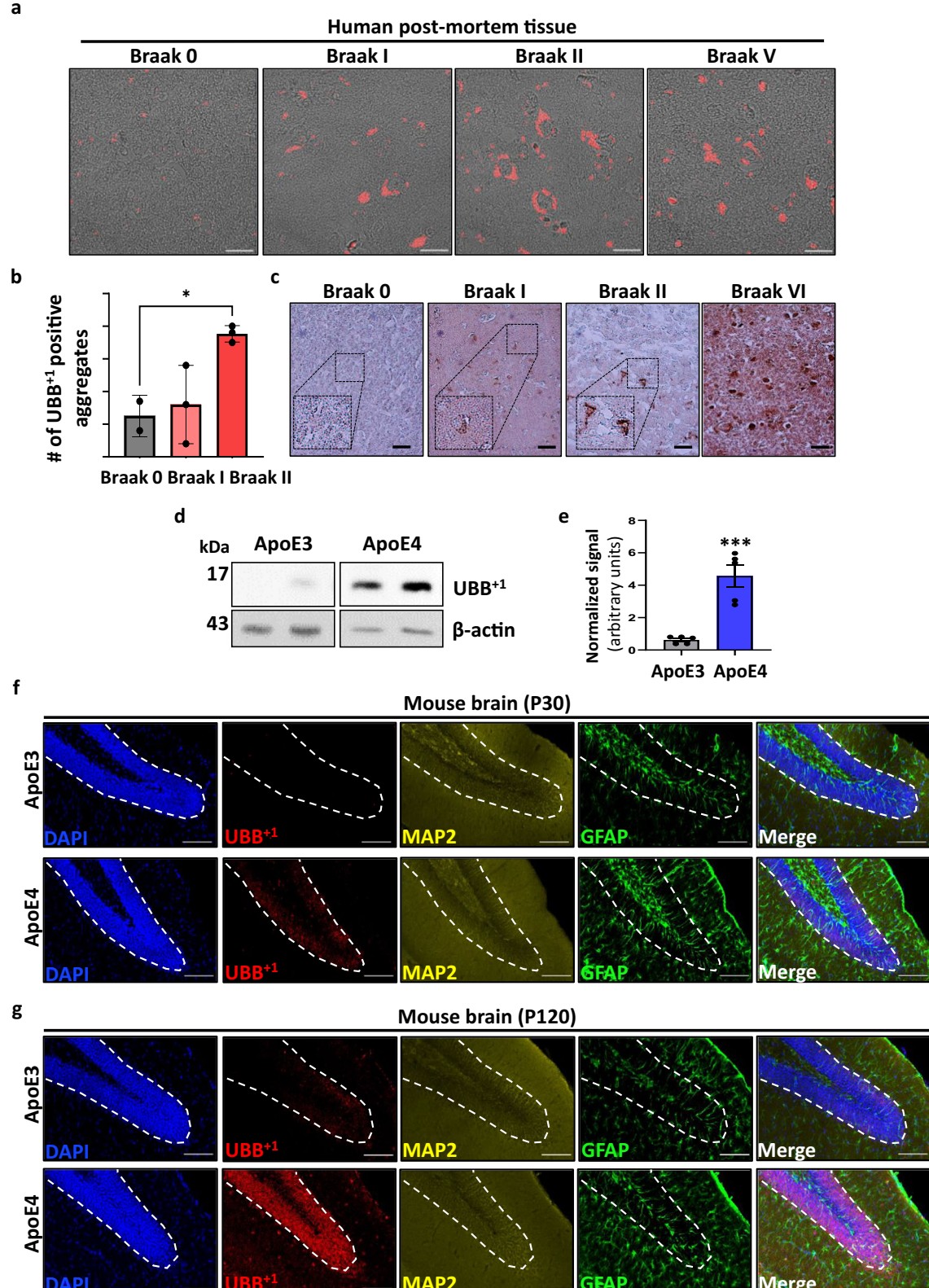

Fig. 2d). Expression of these neuronal markers was similar between the three lines, and no morphological effect was observed on cellular differentiation.

Six weeks post-embedding, expression of UBB$^{+1}$ alone was sufficient to trigger the formation of distinct aggregates that could be immunostained with a specific Aβ antibody, 3D6 (Fig. 2b). Aggregate formation was similar to that observed in the 3D FAD cultures

(Fig. 2b,c). Some earlier studies evaluated the effect of UBB$^{+1}$ on ubiquitin-dependent proteolysis in neuronal cells utilizing a variant of UBB$^{+1}$ with a single site substitution at position 79[30]. In order to link the current study with these published findings, we repeated 3D differentiation expressing the UBB$^{+1(D79S)}$ variant. Aβ staining showed extracellular aggregates 6 weeks post-embedding (Supplementary Fig. 3a,b), and the occurrence of these Aβ aggregates was detected

**Fig. 1 | UBB$^{+1}$ accumulates in the early stages of AD postmortem tissue.**
**a** Immunofluorescence staining of human tissue (ER = hippocampal complex, including entorhinal cortex) with an antibody against UBB$^{+1}$ (red). **b** Quantification of UBB$^{+1}$-positive aggregates counted in different biological samples taken from dentate gyrus (DG) of patients diagnosed at different Braak stages [$n = 3$ biologically independent samples] (Supplementary information Table 1) ($p = 0.038$). **c** Immunohistochemistry staining of human postmortem tissue (ER) with anti-UBB$^{+1}$. **d** Representative immunoblots of proteins isolated from the hippocampus

of Apolipoprotein E (ApoE3/4)-TR mice using an anti-UBB$^{+1}$ antibody. **e** Quantification of (**d**) by densitometry and normalized to β-actin [$n = 5$ mice] ($p = 0.00042$) **f, g** Immunofluorescence staining of hippocampal sections of one (**f**) and 4-month-old (**g**) ApoE (ApoE3/4)-TR mice, showing UBB$^{+1}$ is specifically expressed in neurons (MAP2) and not glial cells (GFAP). P-values were determined by unpaired two-tailed Student's t-test. Error bars represent ± s.d. Images are representative of three independent wells. All experiments were repeated at least twice. Scale bars: 20 μm (**a**), 50 μm (**c**), 100 μm (**f, g**).

even at 3 weeks (Supplementary Fig. 3b). To substantiate the amyloid nature of these aggregates, we stained 6-week-old cultures with the benzothiazole dye Thioflavin S, which specifically marks amyloids (Supplementary Fig. 3c, d). The combined observations all pointed to an increase in the abundance of extracellular aggregates upon UBB$^{+1}$ expression. These aggregates share properties of bona fide Aβ plaques.

Next, we extracted insoluble proteins from 6-week-old 3D cultures grown in a ~4 mm Matrigel layer (referred to herein as "thick"). By immunoblotting the SDS-solubilized sediment with the anti-Aβ antibody, 6E10, we observed Aβ peptides in multimeric forms (Fig. 2d). The pattern distribution of the Aβ species in UBB$^{+1}$ cultures resembled the Aβ species extracted from the 3D FAD cultures, yet differed slightly shifting towards higher MW oligomers. Furthermore, mass spectrometry (MS/MS) analysis performed on this SDS-soluble fraction revealed a four-fold increase in the intensity of APP–derived peptides in the UBB$^{+1}$ sample compared to the control (Supplementary Fig. 3e). Our data indicate that the expression of UBB$^{+1}$ is sufficient to cause accumulation of insoluble Aβ species in cultured human neurons.

Studies with FAD patient-derived human neuronal cells have demonstrated that the ratio of Aβ42 to Aβ40 plays a pathogenic role in AD[31]. The Aβ42/Aβ40 ratio is also used as a biomarker for AD diagnosis[32]. Therefore, we collected conditioned media from 6-week-old 3D cultures to examine the ratio of Aβ42 to Aβ40 secreted from FAD and UBB$^{+1}$ cells using an Aβ ELISA kit. Both FAD and UBB$^{+1}$ cultures showed higher Aβ42/40 ratios as compared to the control tissue (Fig. 2e). Moreover, expression of UBB$^{+1}$ similarly increased the total Aβ42 signal as in FAD cultures (Fig. 2f).

Elevated Aβ42 and/or a high Aβ42/40 ratio have been shown to drive the accumulation of insoluble phosphorylated-tau (p-tau) in a 3D human neural cell culture model of FAD[33]. Therefore, we also investigated whether the expression of UBB$^{+1}$ in 3D neural cultures could lead to tau pathology. IHC with the anti-p-tau antibody AT8 showed that cells in either 8-week-old FAD or UBB$^{+1}$ cultures were positive for p-tau (Fig. 2g,h). The staining pattern was similar to NFT-like structures visualized in other human neuronal models for tauopathies[20,34]. To confirm whether the abundant tau detected in our UBB$^{+1}$-expressing 3D human neuronal cultures also accumulated in aggregates, we extracted proteins from 8-week-old thick layer 3D cultures and performed dot-blot analysis on sarkosyl-insoluble fractions. Immunostaining for p-tau paired helical filaments (PHF-1) and total tau (T46) both resulted in a stronger signal for insoluble material extracted from UBB$^{+1}$ relative to control (Supplementary Fig. 3f). Sarkosyl-soluble fractions isolated from 8-week-old UBB$^{+1}$ cultures were further resolved by SDS-PAGE for immunostaining with PHF-1 and T46 tau antibodies, again showing more intense signals than control and similar to FAD extractions (Fig. 2i). Next, cryogenic transmission electron microscopy (cryo-TEM) was employed to confirm the presence of p-tau in filamentous assemblies. Remarkably, at 12/14 weeks, expression of UBB$^{+1}$ alone was able to induce p-tau positive insoluble aggregates (Fig. 2j, Supplementary Fig. 3g). The overall dimensions and properties of these isolated p-tau-containing deposits are comparable with Alzheimer patient-derived PHFs[35,36] or tau pathology in mouse models[37], although it is worth noting that sarkosyl-insoluble filamentous deposits from

different tauopathies show slightly different conformations at high-resolution imaging[36,38].

The effect of UBB$^{+1}$ appears to be limited to a subset of proteins characteristic of AD pathogenesis since we obtained no evidence of other protein aggregates, such as α-synuclein aggregates, in these cultures (Supplementary Fig. 3h). Moreover, expression of FAD genes or ectopically expressed UBB$^{+1}$ did not alter total ubiquitin levels (Supplementary Fig. 3i).

## UBB$^{+1}$ interacts with UCHL1 and alters APP protein levels

As Aβ plaque formation in the brain is a key pathogenic event in AD progression, we investigated the underlying mechanism of Aβ accumulation following UBB$^{+1}$ expression in our experimental system. First, we asked whether UBB$^{+1}$ cultures secrete Aβ42. To answer this, we co-transfected HEK293FT cells with UBB$^{+1}$-and FAD-expressing plasmids (APP$^{SL}$/PSEN1$^{ΔE9}$) and measured soluble forms of APP (sAPP) in conditioned media. Co-transfection of UBB$^{+1}$ and FAD increased levels of secreted Aβ as compared to FAD alone (Fig. 3a, Supplementary Fig. 4a). Of the various forms of sAPP, Aβ42 is particularly aggregation-prone and may accelerate the formation of neuritic plaques[31,39]. The increase of Aβ42 in the conditioned media of these cells was quantified with an ELISA kit (Supplementary Fig. 4b). The source of Aβ42 is proteolytic processing of the amyloid precursor protein, APP[40]. Cells tightly control APP levels by endosomal-lysosomal processing[41,42], a ubiquitin-dependent process. We confirmed that upon inhibition of lysosomal proteases, APP levels increased in FAD-expressing HEK293FT cells (Supplementary Fig. 4c). Since APP levels are ubiquitin-dependent and UBB$^{+1}$ may interfere with ubiquitin signaling[11,30,43], we overexpressed ubiquitin in this FAD/UBB$^{+1}$ cell line and found that excess of ubiquitin decreases Aβ secretion (Fig. 3b).

We hypothesized that UBB$^{+1}$ could interfere with proteins that associate with ubiquitin domains and in doing so affect ubiquitination processes underlying the secretion of Aβ. To test this hypothesis, we screened for potential UBB$^{+1}$-binding partners by co-immunoprecipitation (co-IP) of UBB$^{+1}$ from HEK293FT cells. MS/MS analysis identified several candidates, the top ubiquitin-related hit was UCHL1 (Supplementary Fig. 4d). Endogenous UCHL1 co-immunoprecipitated with UBB$^{+1}$ (Fig. 3c), and reciprocally, UBB$^{+1}$ co-purified with immunoprecipitated endogenous UCHL1 (Supplementary Fig. 4e). The two also bind directly as recombinant proteins (Supplementary Fig. 4f,g). In addition to UBB$^{+1}$, we observed that UCHL1 could also bind free unconjugated ubiquitin (Supplementary Fig. 4f,g), as has been reported[44]. Indeed, elevated expression of ubiquitin diminished the association of UBB$^{+1}$ with UCHL1 in HEK293FT cells, suggesting that UBB$^{+1}$ and free ubiquitin might compete for binding to the same site in a dose-dependent manner (Fig. 3d).

By competing with ubiquitin for the UCHL1 binding site, UBB$^{+1}$ might interfere with the catalytic activities of UCHL1. Although UCHL1 is generally known as a hydrolase for short ubiquitin extensions[45–47], it has also been shown to catalyze the formation of ubiquitin linkages[48]. The ability of UCHL1 to promote ubiquitination of APP has been attributed to this second role, as a potential ligase[49]. The addition of recombinant UBB$^{+1}$ to UCHL1 enzymatic reactions interfered with both ligase and the hydrolase activities of UCHL1 (Supplementary Fig. 4h, i).

Our finding that UBB$^{+1}$ competes with ubiquitin for binding to UCHL1 may provide a link between UBB$^{+1}$, UCHL1, and APP levels.

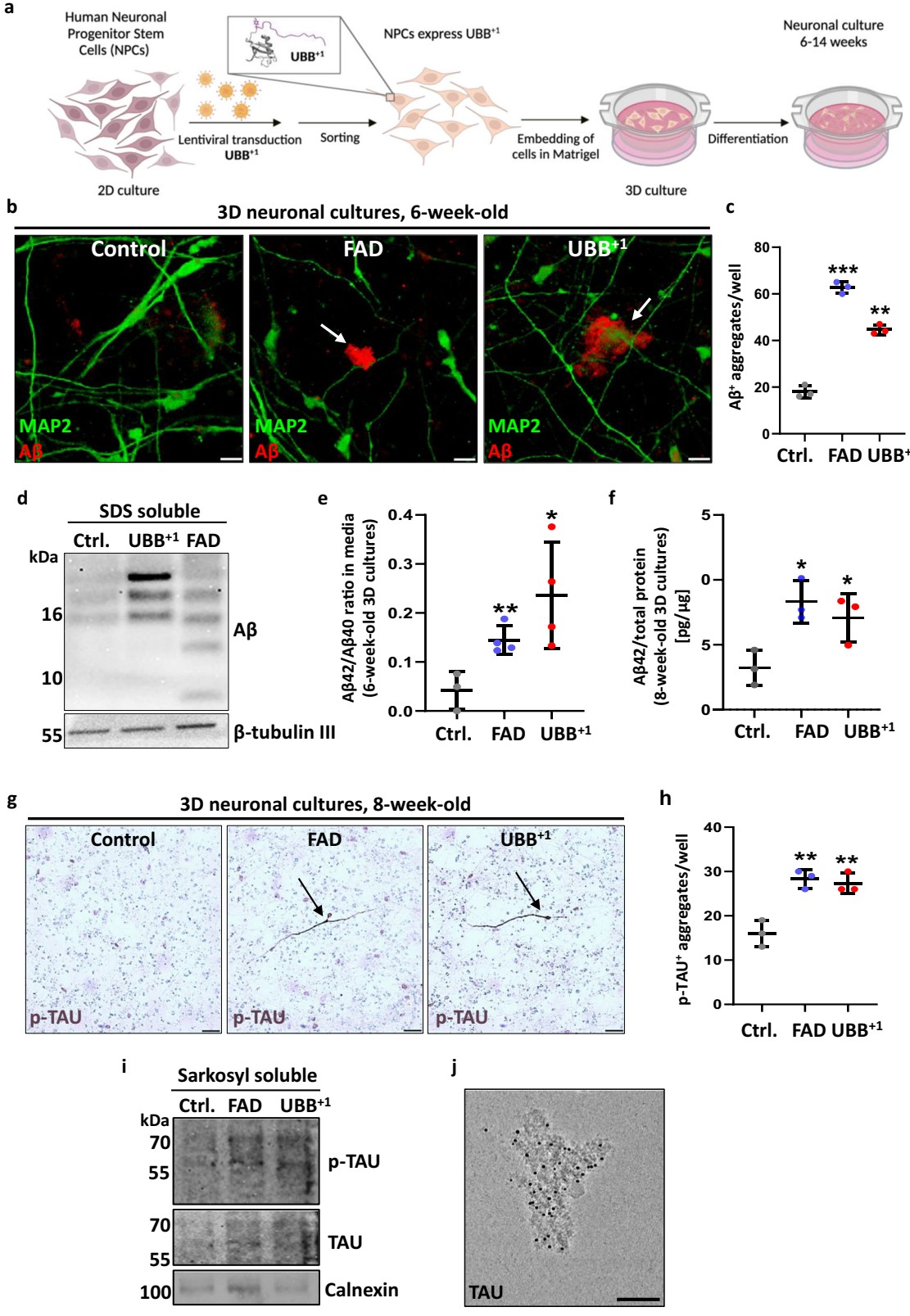

Inhibition of UCHL1 by the compound LDN57444 has been shown to elevate APP levels[49]. Likewise, UBB[+1] expression increased APP protein levels in HEK293FT cells, in alignment with UBB[+1] serving as a potential UCHL1 inhibitor (Fig. 3e). Simultaneous expression of UBB[+1] and LDN57444 treatment did not lead to an additive effect on increasing APP levels (Fig. 3e), potentially implying that UBB[+1] acts on APP via UCHL1. Overexpression of either UCHL1 or ubiquitin depleted APP

levels while increasing its ubiquitination (Fig. 3f; Supplementary Fig. 4j). By contrast UBB[+1] expression resulted in elevated APP levels yet less ubiquitination (Fig. 3f). Previously, neuronal function in AD mice was improved by increasing UCHL1 activity[49,50]. Therefore, we tested whether UCHL1 could also modify UBB[+1]-dependent effects on Aβ[+] aggregates in our 3D model. We transduced 3-week-old 3D UBB[+1] cultures with a UCHL1-overexpressing vector and quantified a

**Fig. 2 | Expression of UBB⁺¹ induces Amyloid beta (Aβ) and phosphorylated tau (p-tau) aggregates in a 3D human neural culture system. a** Schematic of UBB⁺¹ expressing 3D system derived from ReN VM cells. Image was created using BioRender software. **b** Representative immunofluorescence images of 6-week-old differentiated control, APP$^{SL}$-PSNE1$^{ΔE9}$ (FAD) and UBB⁺¹ expressing 3D cultures, showing Aβ-positive-aggregates in white arrows, indicating that they are extracellular deposits. **c** Quantification of the mean total number of extracellular Aβ deposits in 6-week-old control, FAD, and UBB⁺¹ cultures [$n = 3$ biologically independent samples, the whole well was counted] (FAD/ Ctrl. $p = 0.000029$; UBB⁺¹ /Ctrl. $p = 0.00016$). **d** Immunoblot analysis of SDS-solubilized aggregates isolated from 6-week-old -differentiated control, FAD, and UBB⁺¹ cultures, showing Aβ oligomers. β-tubulin III levels were used as loading control. **e** ELISA of Aβ42 and Aβ40 in conditioned media collected from 3D 6-week-old control, FAD, and UBB⁺¹ cultures. Aβ42/Aβ40 ratio is displayed [$n = 3$ biologically independent samples] (FAD/Ctrl. $p = 0.007$; UBB⁺¹ /Ctrl. $p = 0.04$). **f** ELISA of Aβ42 in whole lysates of 3D

8-week-old control, FAD, and UBB⁺¹ cultures. Results are displayed as Aβ42 normalized to total protein [$n = 3$ biologically independent samples] (FAD/ Ctrl. $p = 0.014$; UBB⁺¹ /Ctrl. $p = 0.043$). **g** Representative immunohistochemistry images of 3D 8-week-old FAD and UBB⁺¹ cultures stained with anti- p-tau. Cells displaying increased p-tau staining are marked by black arrowheads. **h** Quantification of the mean number of p-tau positive-stained cells [$n = 3$ biologically independent samples, whole well was counted] (FAD/ Ctrl. $p = 0.004$; UBB⁺¹ /Ctrl. $p = 0.007$). **i** Sarkosyl-soluble fractions extracted from 3D FAD and UBB⁺¹ -8-week-old thick cultures immunoblotted for tau or p-tau. Calnexin levels were used as loading control. **j** Representative image of Cryo-TEM of 14-week-old UBB⁺¹ Sarkosyl-insoluble fraction showing specific binding of tau antibody to aggregates, visualized with 5 nm nano-gold particles. $P$-values were determined by unpaired two-tailed Student's $t$-test. Error bars represent ± s.d. Images are representative of three independent samples. All experiments were repeated at least twice. Scale bars: 20 μm (**b**), 50 μm (**g**), and 100 nm (**j**).

substantially lower number of extracellular Aβ⁺ deposits 1 week later (Fig. 3g,h). We conclude that the effects of UBB⁺¹ on APP levels and Aβ production occur in a UCHL1-dependent manner.

## Silencing UBB⁺¹ expression rescues AD pathology in human FAD neurons

Having demonstrated that a heightened expression of UBB⁺¹ was sufficient to drive AD hallmarks in human neurons, we then asked whether it is also required for FAD phenotypes. Of note, UBB⁺¹ is not a genetic mutation that can be edited back to the wild-type form by gene editing[10]. Thus, we aimed to silence UBB⁺¹ expression specifically, without compromising ubiquitin expression. As a first step, we set out to generate the proper tools and produce efficient RNAi vectors for silencing the UBB⁺¹ transcript. The chosen RNAi construct (shUBB⁺¹) reduced UBB⁺¹ protein levels without interfering with the ubiquitin landscape in the cell extract (Supplementary Fig. 5a, b, f, g). FAD neuronal progenitor cells were then transduced with lentiviruses encoding shRNAs (shUBB⁺¹ or shSCR) and sorted to generate stable lines (Supplementary Fig. 5c). Three weeks post-embedding, UBB⁺¹ immunofluorescence was lower in differentiated FAD+shUBB⁺¹ relative to shSCR control (Supplementary Fig. 5d). Decreased UBB⁺¹ protein levels were also observed by dot-blot analysis of formic acid (FA)-soluble fractions extracted from 6-week-old FAD+shUBB⁺¹ cultures (Supplementary Fig. 5e), once again without affecting total ubiquitin levels (Supplementary Fig. 5f). Specifically, to verify that the UBB transcript is not affected by this shRNA, we extracted RNA from 8-week-old 3D FAD cultures and performed RT-qPCR (Supplementary Fig. 5g). We also note that shUBB⁺¹ had no detectable effect on cleaved caspase 3 in these cultures (Supplementary Fig. 5h,i), indicating no effect on cell viability. Importantly, in 6-week-old FAD cultures, UBB⁺¹ silencing led to a significant decrease of ~60% in extracellular Aβ deposits (Fig. 4a,b). After lysis, dot-blots confirmed a decrease of Aβ levels in sarkosyl-insoluble fractions extracted from FAD+shUBB⁺¹ (Supplementary Fig. 5j). Immunoblotting also verified a decrease in SDS-soluble Aβ species (Fig. 4c). To test the potential effect of shUBB⁺¹ on the Aβ42/Aβ40 ratio, ELISA was performed on conditioned media and on whole culture lysates taken from 6-week-old FAD cultures. In both samples, a significant decrease in Aβ42/Aβ40 ratio was quantified (Fig. 4d,e).

The presence of UBB⁺¹ induced not only Aβ deposits but also p-tau aggregates (Fig. 2). To test the effect of shUBB⁺¹ on tau in the 3D human neural cell culture system[29], FAD cultures were transduced with shUBB⁺¹ (FAD+shUBB⁺¹) and stained with anti-p-tau (AT8) at 8 weeks. The intensity of p-tau staining of FAD+shUBB⁺¹ cells was visibly lower than that of FAD control cells (Fig. 4f). Moreover, the number of positively stained cells decreased as well (Fig. 4g). After lysis, dot-blots confirmed a decrease of aggregated tau and of p-tau in sarkosyl-insoluble fractions extracted from FAD+shUBB⁺¹ using antibodies specific for aggregated tau or phosphorylated forms, respectively

(Supplementary Fig. 5k). Upon solubilization with SDS, depleted levels of tau and p-tau were detected in these fractions using antibodies specific for aggregated tau or phosphorylated forms, respectively (Fig. 4h).

These findings suggest that the AD pathology present in 3D human neuronal systems (FAD and UBB⁺¹) is dependent on the level of UBB⁺¹ and that silencing UBB⁺¹ expression is an effective approach to hinder AD development.

## Discussion

Taken together, our data suggest that expression of UBB⁺¹, in the absence of any known FAD mutations, can recapitulate critical aspects of AD pathology in human neurons. While the pathological manifestation of AD is well-defined, and in many cases, the progression follows a typical pattern, what triggers the onset of sporadic AD is still a mystery. Our results demonstrate that accumulation of UBB⁺¹ in the brain is an early event in AD pathogenesis and that increased expression of UBB⁺¹ is sufficient to induce AD-like pathology (i.e., Aβ-containing plaques and phosphorylated tau aggregates) in human neurons. As a disease characterized by the accumulation of protein aggregates, it is reasonable to propose a pivotal role for the ubiquitin system in AD pathogenesis[5,10,51,52]. Enhanced proteasome activity protects APP-overexpressing mouse and fly models from AD-like pathology[9], and conversely, elevated USP11-driven deubiquitination of tau increases pathological tau aggregation[51]; both imply that ubiquitination can mitigate AD pathology. Likewise, interfering with the normal function of UCHL1 causes various abnormalities in neurons[50].

Mechanistically, we now show that UBB⁺¹ competes with ubiquitin for binding to UCHL1, resulting in reduced ubiquitination of APP, elevated APP, increased secretion of Aβ peptides, and Aβ plaque formation. Interestingly, ubiquitinated APP appears to undergo mainly lysosomal degradation[49]. For instance, recent findings showed that the buildup of APP species within enlarged de-acidified autolysosomes is an early feature of AD neurons that precedes the formation of Aβ plaques[53]. The precise molecular basis of lysosome dysfunction in AD is unclear, but impaired ubiquitin signaling is a potential contributor since trafficking to the lysosome is largely ubiquitin-dependent[6,54]. Merely harboring an extra copy of the APP gene attributes to early-onset forms of AD[55,56]. Thus, conditions that increase APP have the potential to increase the risk of AD. By pinpointing the interference of UBB⁺¹ with a specific APP-interacting target, UCHL1[49], our current findings highlight a link between ubiquitin signaling and the generation of Aβ.

Tau aggregates in the UBB⁺¹-expressing neural cultures are biochemically and ultrastructurally similar to NFTs present in AD brains[36]. NFTs are possibly the result of an increased Aβ42/40 ratio in tissues[31,33], a phenomenon observed also upon UBB⁺¹ expression in the current study. The exact nature of this relationship remains to be investigated

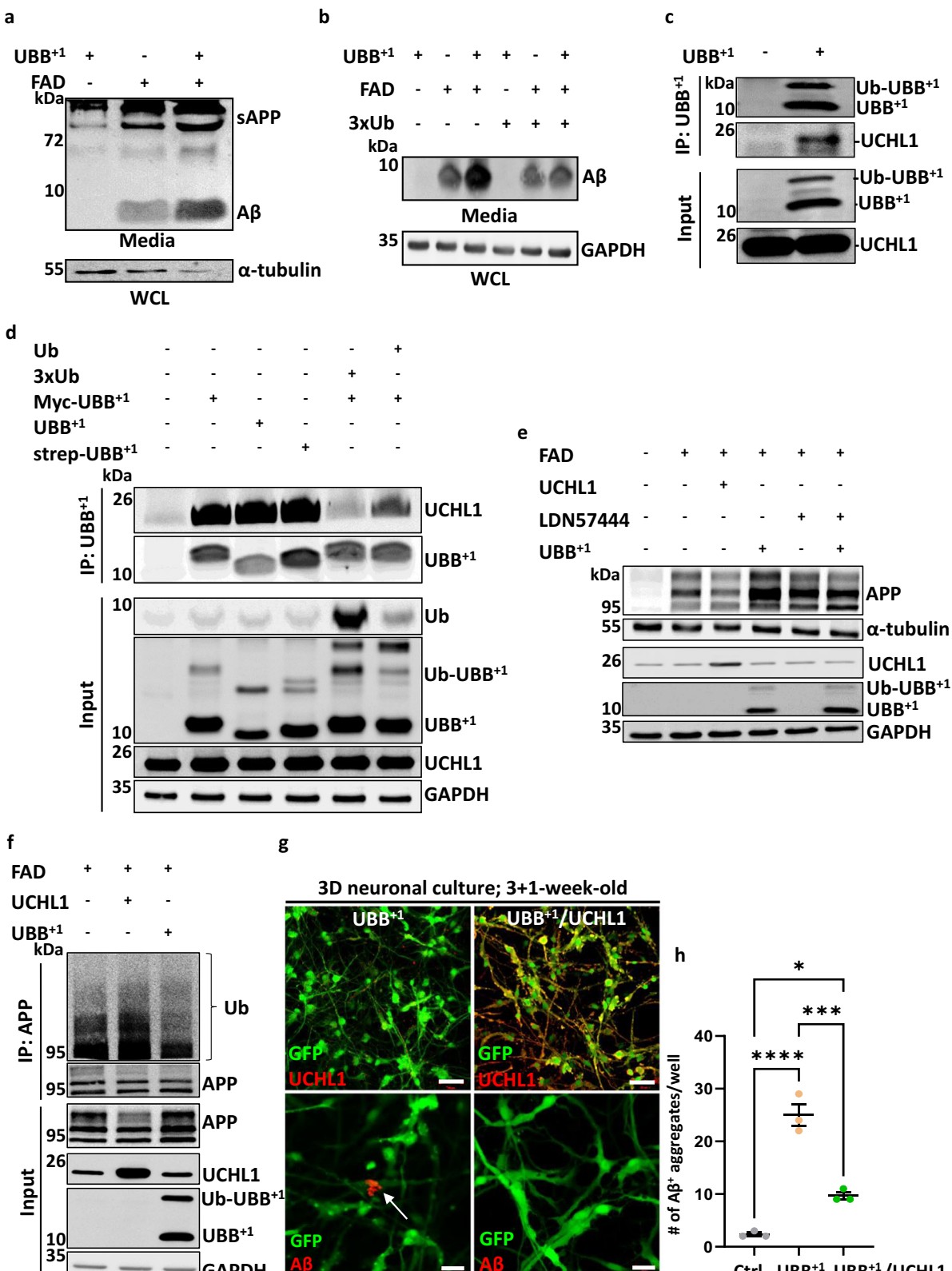

in more detail as UBB[+1] may also induce tau aggregates through an Aβ-independent mechanism. Regardless, in neurons carrying FAD mutations (APP[SL]/PSEN1[ΔE9]), this AD-like pathology could be rescued by silencing the expression of endogenous UBB[+1] using an RNAi approach. This improvement hints at a causal link between UBB[+1] and cellular hallmarks of AD and supports a central and crucial role for defective ubiquitin signaling in the development of the disease.

In conclusion, the results described herein indicate that altered ubiquitin signaling is an important component of AD pathogenesis and that ubiquitin signaling could be a therapeutically actionable target to ameliorate AD pathology. With the FDA modernization act 2.0 "Animal Testing Alternatives" approving new in vitro testing methods in stem cell-derived cells and tissues[57], the present study provides a proof-of-concept for generating human-relevant "nonclinical" models of SAD.

**Fig. 3 | UBB$^{+1}$ binds UCH-L1 and increases APP protein levels. a** Representative immunoblot of Aβ from conditioned media collected from HEK293FT cells transfected with either APP$^{SL}$-PSNE1$^{ΔE9}$ (FAD), UBB$^{+1}$, or both (top panel). α-tubulin immunoblot in whole cell lysate (WCL) from the same culture (bottom panel). **b** Representative immunoblot of Aβ from conditioned media collected from HEK293FT cells expressing FAD or UBB$^{+1}$ or both, with or without overexpression of ubiquitin (3xUb). GAPDH immunoblot in WCL from the same culture (bottom panel). **c** Co-IP was performed with an anti-UBB$^{+1}$ antibody from HEK293FT WCL expressing UBB$^{+1}$ or an empty vector, immunoblotted for UCHL-1 and UBB$^{+1}$. Note the higher migrating form of UBB$^{+1}$ that has been confirmed to be a ubiquitinated form of UBB$^{+1}$. **d** Co-immunoprecipitation of UBB$^{+1}$ from HEK293FT cells expressing one of three UBB$^{+1}$ constructs - UBB$^{+1}$, STREP-UBB$^{+1}$, or Myc-UBB$^{+1}$ - and co-transfected with either one copy of ubiquitin or 3 copies of ubiquitin (3xUb). **e** Immunoblot of APP in HEK293FT cells transfected with FAD and UCHL1 or UBB$^{+1}$

as noted. Some samples were treated with the UCHL1-specific chemical inhibitor LDN57444. **f** Ubiquitin immunoblot of APP IP from HEK293FT cells transfected with FAD and either UBB$^{+1}$ or UCHL1 as noted. WCL was immunoblotted as indicated. GAPDH was used as a loading control. All blots in **a–f** were repeated at least twice. **g** Representative immunofluorescence images of UCH-L1 expression (top panel) and Aβ staining (3D6; white arrows, bottom panel) in 4-week-old 3D UBB$^{+1}$ and UBB$^{+1}$/UCH-L1 cultures. At 3 weeks, half of the UBB$^{+1}$ cultures were transduced with UCHL1 lentiviral particles (right) and the other half with mock particles (left). **h** Quantification of the mean total number of extracellular Aβ+ deposits per well in UBB$^{+1}$ and UBB$^{+1}$/UCH-L1 cultures [$n = 3$ independent wells, whole well counted] (Ctrl./UBB$^{+1}$ + UCHL1 $p = 0.016$; UBB$^{+1}$ /UBB$^{+1}$ + UCH-L1 $p = 0.0004$ Ctrl./UBB$^{+1}$ $p = 0.0001$). *P*-values were determined by one way ANOVA test. Error bars represent ± s.d. Scale bars: 50 μm (**g**, top panel), 20 μm (**g**, bottom panel).

Future studies could improve the complexity of the UBB$^{+1}$-expressing 3D cultures, by incorporating additional cell types that enable the study of relevant disease processes, such as neuroinflammation[58], *APOE4*[23], or other factors that are suspected to augment or ameliorate the interference that UBB$^{+1}$ has on ubiquitin-dependent protein homeostasis. In parallel, other assaults on the ubiquitin system could also be tested. It would be interesting to interrogate the roles of different factors implicated in AD using such platforms to elucidate the sequential steps leading to AD and to screen effectively for new therapeutic targets that slow down, stop, or reverse AD progression.

## Methods

All applicable international, national, and/or institutional guidelines for the care and use of previously published animal or human samples were followed (Tel Aviv University for mice samples and MST Hospital Group in Enschede for human samples).

### Cell lines and media

HEK293FT cells were purchased from ATTC (Manassas, VA) and maintained in Dulbecco's Modified Eagle Medium-DMEM (Biological Industries) supplemented with 10% FBS (Biological Industries) L-glutamine, sodium pyruvate, sodium bicarbonate and penicillin/streptomycin (Biological Industries; 1:100). For transfection, cells were plated onto poly-L-lysine (Sigma-Aldrich) and transfected using polyethyleneimine, branched (MERK). SK-N-SH cell line was purchased from ATTC (Manassas, VA) and maintained in Eagle's Minimum Essential Medium (Biological Industries) supplemented with 10% FBS (Biological Industries), L-glutamine, and penicillin/streptomycin (Biological Industries; 1:100). VM Human Neural Progenitor Cell Line (ReN) were purchased from EMD Millipore (Billerica, MA, USA) and maintained in DMEM/F12 (Biological Industries) media supplemented with 2 μg/ml heparin (StemCell Technologies, Vancouver, Canada), 2% (v/v) B27 neural supplement (Life Technologies, Grand Island, NY, USA), 20 ng/ml EGF (Peprotech), 20 ng/ml bFGF (Peprotech) and 1% (v/v) penicillin/streptomycin/amphotericin-b solution (Biological Industries). For 3D neuronal/glial differentiation, cells in Matrigel (Corning) were plated on either 24-well glass-bottom plates (Greiner-bio-one), 35 mm glass-bottom plates (ibidi), or cell culture inserts, 0.4 μm pore size (Greiner-bio-one) and companion plates, 24 well, for cell culture inserts (Greiner-bio-one). The cells were differentiated with DMEM/F12 media supplemented with 2 μg/ml heparin, 2% (v/v) B27 neural supplement, and 1% (v/v) penicillin/streptomycin/amphotericin-b solution without growth factors. Half of the volume of the differentiation medium was replaced every 4 d for a duration of 3–14 weeks.

### DNA constructs and viral packaging

Lentiviral polycistronic CSCW vectors including CSCW -GFP, CSCW-mCherry, and CSCW-APP-IRES-PSEN1(ΔE9)-IRES-mCherry, encoding full-length human β-amyloid precursor protein (APP$_{695}$) with the V717I (London) and K670N/M671L (Swedish) mutations and the human

presenilin 1 (PSEN1(ΔE9)) were kindly donated by Rudolph E. Tanzi and Doo Yeon Kim (Harvard University)[29]."WT" UBB$^{+1}$ vector was created by inserting a UBB$^{+1}$ encoding gBLOCK DNA (IDT Technology) into CSCW-mCherry cut by xho1-Bstb1. mycUBB$^{+1D78S}$ vector was kindly donated by Nico Dantuma (Karolinska Institute)[30] and cloned into CSCW–GFP and CSCW–mCherry backbones using XhoI and BstbI.

To generate lentiviral vectors for RNA silencing, we restricted *LV-GFP* lentiviral vector (Addgene plasmid #25999) with RsrII and EcoRI enzymes. shRNA oligos were annealed and cloned into *LV-GFP* cleaved vector. Sequences are as follows:

shUBB$^{+1}$(6) F
5'GTC TCT GAG AGG GTA TGC AGA TCT CTC GAG AGA TCT GCA TAC CCT CTC AGA TTT TTG-3';
shUBB$^{+1}$(6) R
5'AAT TCA AAA ATC TGA GAG GGT ATG CAG ATC TCT CGA GAG ATC TGC ATA CCC TCT CAG A-3'.
Scrambled control forward sequence:
5'GTCACAGAGAGATAGCGCGTGTGTCTCGAGACACACGCGCTAT CTCTCTGTTTTTTG-3';
Reverse sequence:
5'AATTCAAAAAACAGAGAGATAGCGCGTGTGTCTCGAGACA-CACGCGCTATCTCT CTGT-3'.

Plasmids were transfected into HEK293FT cells and lentiviral particles were collected and concentrated using centricons (Merck Millipore), for lentiviral infection of SK-N-SH and ReN VM cells using polybrene. For recombinant expression and purification of GST and His tagged proteins, UCHL1 was inserted in the pGEX backbone using BamHI and XhoI, HIS-Thrombin-UBB$^{+1}$ / ubiquitin was inserted in pQE30 using BamHI, and SalI.

### FACS enrichment of the transduced cells

Briefly, infected ReN cells were washed with PBS and incubated with Accutase (Merck Millipore) for 5 min. The cell pellets were resuspended in PBS supplemented with 2% serum replacement solution and 2% B27 (Life Technologies), and then passed through a cell strainer filter (40μm Nylon, BD Biosciences). Cell concentrations were adjusted to $3 \times 10^6$ cells per ml and sorted using the FACSAriaIIIu cell sorter. Dead cells, debris, and doublets were gated out according to FSC and SSC properties. Sorting gates were set to select cells with high expression (top 2%) mCherry. After sorting, cells were seeded immediately and maintained in normal proliferation media.

### 3D cell cultures and differentiation

3D cultures were produced and maintained as described[29]. Briefly, for thin-layer 3D cultures, Matrigel stock solution was diluted with ice-cold differentiation medium (1:11 dilution ratio) and then vortexed with the cell pellets. The final cell concentration for the mixture was ~$1 \times 10^7$ cells per ml. Plates were incubated for 24 h at 37 °C to form thin-layer 3D gels and the media was added the next day. 3D-plated cultures were differentiated for 3–14 weeks and media was replaced every 4 days. For

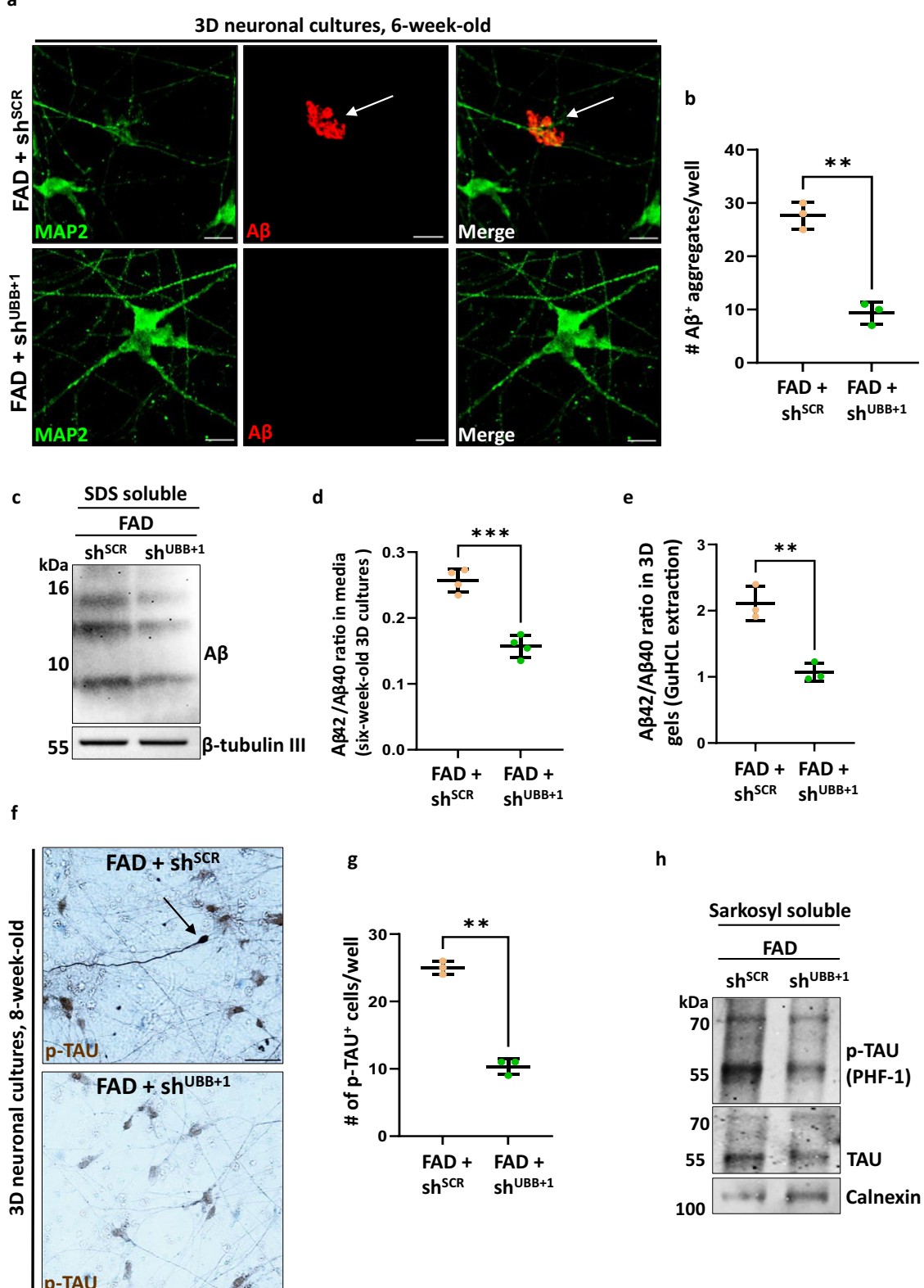

thick-layer 3D cultures, Matrigel solution was diluted with the same volume of ice-cold differentiation medium (1:2 dilution ratio) and vortexed with ReN cell pellets for 20 s. The final cell concentration for the mixture was $1 \times 10^7$ cells per ml. After 24 h incubation at 37 °C, differentiation media was added and the cultures were maintained for 3–14 weeks.

**Immunofluorescence staining**

For SK-N-SH cells, samples were fixed with 4% PFA at RT for 20 min. Fixed cells were then permeabilized with 0.1% Triton X-100 and blocked in 10% BSA at RT for 1 h. Washing was performed with a TBS buffer containing 0.2% (v/v) Tween-20 (TBST). Cells were incubated with primary antibodies in 2% BSA at 4 °C O/N with gentle rocking.

**Fig. 4 | Silencing UBB[+1] expression decreases Aβ and p-tau aggregates in a 3D human neuronal culture. a** Immunofluorescence of Aβ deposits in 6-week- old FAD+sh[SCR] or FAD+sh[shUBB+1] (green, MAP2; red, 3D6; arrowheads, extracellular Aβ deposits). **b** Quantification of extracellular Aβ deposits in FAD+sh[SCR] or FAD+sh[shUBB+1] cultures [*n* = 3 biologically independent samples, the whole well was counted] (*p* = 0.0006)**c** Immunoblot of Aβ aggregates from 6-week-old FAD+sh[SCR] or FAD+sh[shUBB+1] cultures. **d** Quantification of Aβ42/Aβ40 by ELISA of conditioned media from 6-week-old FAD+sh[SCR] or FAD+sh[shUBB+1] cultures [*n* = 4 biologically independent samples] (*p* = 0.00018). **e** Quantification of Aβ42/Aβ40 by ELISA of lysates from 6-week-old FAD+sh[SCR] or FAD+sh[shUBB+1] cultures [*n* = 4 biologically independent samples] (*p* = 0.0373). **f** Immunohistochemistry of 8-week-old FAD+sh[SCR] or FAD+sh[shUBB+1] cultures showing p-tau staining (brown, p-tau; arrows indicate cells with high levels of p-tau). **g** Quantification of p-tau deposits in 8-week-old FAD[SCR] or FAD[shUBB+1] cultures. [*n* = 3 biologically independent samples, the whole well was counted] (*p* = 0.00007). **h** Immunoblot of 8-week-old FAD+sh[SCR] or FAD+sh[shUBB+1] cultures using anti-p-tau, and anti-tau, Calnexin was used as housekeeping gene. *P*-values were determined by unpaired two-tailed Student's *t*-test where. Error bars represent ± s.d. Images are representative of three independent wells. All experiments were repeated at least twice. Scale bars: 20 µm (**a**), 50 µm (**f**).

After washing three times with TBST, the cells were then incubated with AlexaFluor secondary antibodies for 1 h at RT. After washing three times with TBST, flouromountG (ThermoFisher Scientific) was added to the samples. Immunofluorescence staining for 3D-cultured ReN cells was performed as published[29], utilizing the following antibodies: 6E10 anti-amyloid-β antibody (1:100, Biolegend, Cat. no. 803015); 3D6 anti-amyloid-β antibody (1:400, Creative Biolabs, Cat. no. PABL-011), anti-GFAP antibody (1:2,000, DAKO, Cat. no. Z0334); anti-tyrosine hydroxylase (1:100, Cell Signaling Technology, Cat. no. 2792); anti-NR2B (1:100, Antibodies Incorporated, Cat. no. N/59/36). AlexaFluor 488/546 anti-mouse, and -rabbit secondary antibodies (1:400, Life Technologies, Cat. no. A-11011, A-11003, A-11034, A-11035). For UBB[+1] staining; thin-layer 3D cultures were fixed with 4% PFA at room temperature for 24 h. The fixed cells were then permeabilized with TBS buffer containing 0.1% (v/v) Tween-20 (TBST) including 0.1% Triton X-100 at RT for 1 h. Next, samples were blocked by incubating with a blocking solution containing 5% goat serum, 1% BSA, 2% gelatin, and 0.3% triton 100x in PBS at RT for 5 h. After washing briefly with TBST, the 3D cultures were incubated with anti UBB[+1] (1:500, custom-made by Sigma Aldrich for our lab using the C-terminus 20aa sequence YADLREDPDRQDHHPGSGAQ[59,10]) in the blocking solution at RT O/N. After washing five times with TBST for 10 mins, the cells were then incubated with AlexaFluor secondary antibodies for 5 h at RT and washed with TBST for at least 2 h at RT. For *ApoE*-TR mice, immunostaining was performed as previously described[27], utilizing the following antibodies: anti UBB[+1] (1:500), anti-GFAP antibody (1:200, SANTA CRUZ biotechnology, Cat. no. sc-33673); and anti-MAP2 (1:1000, Abcam, Cat. no ab5392). Fluorescence images were captured by a Zeiss LSM880 confocal microscope.

### Thioflavin S staining

Thioflavin S staining protocol was used for staining thin 3D cultures. Samples were fixed, permeabilized, and blocked as described for immunofluorescence[29]. Next, cultures were incubated with 0.02% Thioflavin S (Sigma-Aldrich) in 50% ethanol for 10 min at RT[60]. After briefly rinsing with distilled water three times, samples were washed several times in 50% and 70% Ethanol O/N at RT, samples were then washed twice with DDW, and fluoromountG was added.

### Immunohistochemistry

For IHC, thin-layer 3D cultures were permeabilized and blocked by incubating with a blocking solution at 4 °C for 12 h. To block endogenous peroxidase activities, the cultures were incubated with 0.3% (v/v) H₂O₂ solution in TBS for 5 min at room temperature, washed with TBST five times, and incubated with the blocking solution for 12 h at 4 °C. After incubating with the primary antibody solutions for 24 h at 4 °C, the cultures were washed five times with TBST and then incubated with ImmPRESS anti-mouse Ig (ImmPRESS Peroxidase Polymer Detection Kit, Vector Laboratories, Burlingame, CA, USA) for 30 min. The cultures were washed five times for 10 min each with TBST and developed by using an ImmPACT DAB Peroxidase Substrate kit (Vector Laboratories). The following antibodies and dilution rates were used: AT8 anti-p-tau antibody (1:30, Thermo Scientific, Cat. no. MN1020).

### IHC and IF on human sections

Paraffin-embedded tissues were kindly donated by the MST Hospital Group in Enschede headed by Dr. Robert de Vos. Samples were deparaffinized, followed by incubating 100% formic acid for improved penetration of the antibody. For IHC, to block endogenous peroxidase activities, the samples were incubated with 3% (v/v) H2O2 solution in TBS for 10 min at room temperature, washed with TBST five times, and blocked by incubating with blocking solution (5% Goat serum, 0.2% triton in PBS) for 2 h. After incubating with the primary antibody solutions for 24 h at 4 °C, the samples were washed six times with TBST and then incubated with ImmPRESS anti-Rabbit Ig (ImmPRESS Peroxidase Polymer Detection Kit, Vector Laboratories, Burlingame, CA, USA) for 1 h. The samples were washed six times for 20 min each with TBST and developed by using an ImmPACT DAB Peroxidase Substrate kit (Vector Laboratories). The following antibodies and dilution rates were used: UBB[+1] (1:500, custom-made by Sigma Aldrich for our lab using the C-terminus 20aa sequence). For IF staining, the samples were blocked by incubating with a blocking solution (5% Goat serum, 0.2% triton in PBS) for 2 h. After incubating with the primary antibody solutions for 24 h at 4 °C, the samples were washed six times with TBST and then incubated with AlexaFluor secondary antibodies for 1 h. The samples were washed six times for 20 min each with TBST. The following antibodies and dilution rates were used: UBB[+1] (1:500, custom-made by Sigma Aldrich for our lab using the C-terminus 20aa sequence). Fluorescence images were captured by a Zeiss LSM880 confocal microscope.

### Differential extraction of thick 3D cultures

Six-week-old thick-layer 3D cultures were collected, saved at -80°C until analysis, and extracted as described[29] with minor modifications: Cell pellets were thawed on ice, and 300 µl of cell recovery solution (CORNING, cat. no. 354253) was added, prior to extraction, in order to retrieve them from Matrigel. Cells were incubated on ice for 30 mins, followed by centrifugation at 8,000xg for 10 mins at 4 °C. The supernatant was discarded and pellets were treated again as above, using 100 µl of cell recovery solution. Supernatant was discarded and pellets were washed with 500 µl of cold PBS, centrifuged for 10 mins at 4 °C. β-tubulin levels of SDS-soluble fractions or TBS fractions were used to normalize the total protein levels in SDS and formic acid fractions.

### Transmission electron microscopy (TEM) and Cryo-TEM

Cryogenic transmission electron microscopy (cryo-TEM) and room-temperature TEM for negative staining were both performed by a FEI T12 G² electron microscope, operated at 120 kV. Images were recorded digitally by a Gatan US 1000 CCD camera (Tecnai T12 G²), using the DigitalMicrograph® software. Immunogold staining: Immunogold staining of sarkosyl-insoluble samples for TEM analysis was performed as previously described[29] with minor changes as follows: sarkosyl-insoluble fractions were resuspended in 100 µl of PBS. Samples were incubated with a primary antibody (mouse anti-tau46, 1:25, Cell signaling, Cat. no. 4019 S) or AT8, (1:30, Thermo Scientific, Cat. no. MN1020) for 1 h at RT, next, a secondary antibody was added (goat

anti-mouse IgG 5 nm gold, 1:15, Sigma Aldrich Cat. no. G7527) and incubated for 1 h. Negative staining specimen preparation for TEM: Negatively stained specimens were prepared on 200-mesh copper grids coated with carbon type-B continuous film. Before specimen preparation, the grids were plasma etched in a PELCO EasiGlow glow-discharger (Ted Pella Inc., Redding, CA) to increase their hydrophilicity and clean their surface. A grid was placed on the surface of a 15-μl drop of sample for 2 min, with the coated side facing the drop. The grid was then placed on a 15-μl drop of 2% Uranyl Acetate stain for a 2-min incubation.

Cryo-TEM specimen preparation: Cryo-TEM specimens were prepared in a 25 °C temperature-controlled chamber with humidity at saturation vitrification system. A drop of the solution was placed on a carbon-coated perforated polymer film, supported on a 200 mesh TEM grid, and mounted on tweezers. The drop was turned into a thin film and the grid was then plunged quickly into liquid ethane at its freezing point (-183 °C). Before specimen preparation, grids were plasma etched in a PELCO EasiGlow glow-discharger (Ted Pella Inc., Redding, CA) to increase hydrophilicity. Following preparation, specimens were transferred under cryogenic conditions into a Gatan 626.6 cryo-holder and equilibrated below -170 °C for imaging in the TEM.

### Immunoprecipitation and affinity purification

To test for binding partners of $UBB^{+1}$ or UCH-L1, co-immunoprecipitation was performed in HEK293FT cells. Cells were transfected with $UBB^{+1}$ expression vector and lysed after 24 h. Cells were resuspended in an equal volume of lysis buffer (20 mM HEPES, pH s7.4, 150 mM NaCl, 10% glycerol, 1% Triton X-100, 1 mM EGTA) containing protease inhibitors (Roche). Lysate was collected into a fresh, ice-cold tube and incubated on ice for 30 min, and centrifuged at 11,000x g, at 4 °C for 10 min. 15μl of each lysate was put aside for total protein determination. To identify UCH-L1-binding partners, lysates were incubated with either anti-UCH-L1 (Abcam, Cat. no. ab8189) or anti-$UBB^{+1}$ antibodies overnight with slow rotation at 4 °C. The next day 40μl A/G beads (Santa Cruz Biotechnology) were added to each lysate and incubated for at least 2 h with slow rotation at 4 °C. Beads were washed 4 times with 1 ml lysis buffer and finally, 40μl of Protein Sample Buffer concentrated X2 was added to the beads and heated at 95 °C for 10 min. To identify APP ubiquitination, immunoprecipitation of APP C-terminus was performed as described previously[49], utilizing anti-C-terminus APP antibody (MERK, Cat. no. 751-770) and A/G beads, elution fraction was resolved by SDS-PAGE and immunoblotted with anti-ubiquitin antibody (Dako, Cat. no. Z0458). Pull-down by affinity co-purification of recombinant proteins was performed by incubation of 50 μg GST-UCH-L1 with 50 μg 6XHIS-$UBB^{+1}$/Ubiquitin in 500 μl TBS for 2 h at 25˚C and purified over Ni-NTA (ABT) according to manufacturer's protocols. A reciprocal experiment was performed by incubating 50 μg 6XHIS-$UBB^{+1}$/Ubiquitin in 500 μl TBS with immobilized GST-UCH-L1 on GSH-sepharose (ABT) and eluting according to the manufacturer's protocol. Affinity purification of STREP- $UBB^{+1}$ was performed according to the purification protocol (Strep-Tactin XT Spin Column, IBD Solutions For Life Sciences).

### Extraction of amyloid-β from medium

HEK293FT cells were transfected with the relevant vectors in a serum-free medium. 24 h after transfection, media was collected and methanol precipitation was performed as follows: Four volumes of 100% ice-cold methanol were added to each sample, vortexed, and incubated at −80 °C O/N. The next day, samples were centrifuged at 4,000 g, 4 °C, for 10 min, and the supernatant was removed carefully. Samples were washed twice with 70% ice-cold methanol and the pellet was air-dried for ~20 min. Pellets were resuspended in protein loading dye x1 and run on a 16% tris-tricine gel.

### Ligase assay

Ligase assay was performed according to a published approach[48]. Specifically, we purified recombinant GST-UCHL1, His$_6$-ubiquitin, and His$_6$-$UBB^{+1}$. Ligation of Ub-AMC was performed by mixing UCH-L1 (1 μM): Ub-AMC (3 μM; Boston Biochem) at 1:3 in reaction buffer containing 50 mM TRIS pH 7.4 and 5 mM DTT at 37 °C for 3 h with slow rotation. The reaction was repeated with the addition of a 50X fold of either Ubiquitin (50 μM) or $UBB^{+1}$ (50 μM). Loading buffer was added to the samples, and samples were loaded to 16% tris-tricine gel and transferred to nitrocellulose membrane for immunoblotting against Ubiquitin (1:2000, DAKO, Cat. no. Z0458). Before incubation, 15 μl of each sample was taken for analysis of hydrolysis of the AMC conjugate by measuring the fluorescence of released AMC by spectrophotometer (CLARIO, BMG LABTECH) similar to a published approach[61].

### Western blot analysis

Proteins were resolved on 8%, 10%, 15%, or 4-20% tris-glycine gels, and for amyloid-β or Ub immunoblotting 16% tris-tricine gel was used[62], transferred to nitrocellulose membranes (GE Whatman, 0.2 μm Pore Size). Western blot images were visualized by enhanced chemiluminescence (ECL). The images were captured by using an ImageQuant™ LAS 4000 imaging system (GE Healthcare) and quantitated by Image Studio™ Lite (LI-COR Biosciences). Primary antibodies were used at the following dilutions: 6E10 anti-amyloid-β (1:1000, Biolegend, Cat. no. 803015); anti-PSEN1 (1:1,000, Cell Signaling Technology, Cat. no. 5643); anti-β-tubulin (1:1,000, Abcam, Cat. no. AB24629); anti-$UBB^{+1}$ (1:1000, custom made by Sigma for our lab), anti-ubiquitin (1:2000, DAKO, Cat. no. Z0458), anti-UCH-L1 (1:1000, Abcam, Cat. no. ab8189), anti-GAPDH (1:10,000, Sigma, Cat. no. G9545), anti-β-actin (1:10,000, Santa Cruz, Cat. no. sc-47778).

### Dot blot analysis

Dot blot was performed as previously described[29]. Primary antibodies were used at the following dilutions: 6E10 anti-amyloid-β (1:1000, Biolegend, Cat. no. 803015); anti-$UBB^{+1}$ (1:1000, custom made by Sigma for our lab), anti-MC1 (1:1000, kindly donated by P. Davies) and anti-PHF-1 (1:1000, kindly donated by P. Davies).

### Mouse tissue

*ApoE3*-TR and *ApoE4*-TR mice were purchased from Taconic (Germantown, NY). In these mice, the endogenous mouse *ApoE* was replaced by either human *APOE3* or *APOE4*. In order to minimize possible genetic drift between the apoE4 and apoE3 mice, which were offspring of apoE4 and apoE3 homozygous mouse colonies generated by Taconic around 2001, the mice were back-crossed to wild-type C57BL/6 J mice (2BL/ 610; Harlan Laboratories) for 10 generations and were homozygous for the *ApoE3* (3/3) or *ApoE4* (4/4) alleles. The *ApoE* genotype of the mice was confirmed by PCR analysis. The present experiments were performed with apoE3 and apoE4 homozygous mice on an α synuclein -/- background in which the apoE4-driven synaptic and AD-related phenotype is more pronounced. Mice were anesthetized with ketamine and xylazine and perfused transcardially with phosphate-buffered saline. Brains were then removed and halved, and each hemisphere was further processed for either histologic or biochemical analysis. All experiments were performed on 4-month-old male mice and approved by the Tel Aviv University Animal Care Committee. Every effort was made to reduce animal stress and minimize animal usage[27].

### MS/MS analysis of Aβ peptides in 3D culture

Samples were digested by trypsin (Promega, Cat. No V5111), analyzed by LC-MS/MS on the HFX mass spectrometer (Thermo), and identified by Discoverer software version 1.4 vs the human Uniprot database and against decoy databases (to determine the false discovery rate (FDR)) and vs the specific sequences of the interested proteins using the

Mascot and Sequest search engine. Semi-quantitation was done by calculating the peak area of each peptide.

## Quantitative analysis of Aβ levels

A multi-array ELISA assay kit simultaneously measured levels of Aβ40 and Aβ42 in media (DAB140B, DAB142 kit, R&D Systems). To analyze accumulated Aβ levels in 3D gels, cells were 3D-differentiated for 6 weeks. The extraction was done as described[63,64], and the measurement was done according to the manufacturer's protocol.

## RNA extraction, reverse transcription, and real-time PCR

RNA was isolated using TRIzol (Sigma), and 100 ng of RNA was subjected to cDNA synthesis (PCR Biosystems). Real-time PCR was carried out using the qPCRBIO SyGreen Blue Mix Hi-ROX (PCR Biosystems), with gene-specific primers GAPDH: GGAGCGAGATCCCTCCAAAAT and GGCTGTTGTCATACTTCTCATGG, Ubiquitin B: GGTCCTGCGTC TGAGAGGT and GGCCTTCACATTTTCGATGGT. Amplicon levels were analyzed in triplicates and quantitated relative to a standard curve comprising cDNA and values normalized to housekeeping gene levels (GAPDH). Reactions were as follows: 3 min at 95 °C, then 40 cycles of 10 s at 95 °C and 30 s at 60 °C with the addition of melt curve step: 10 s at 95 °C, and increments of 0.5 °C every 5 s between 65 °C to 95 °C.

## Statistics

All statistical analyses were performed either using a two-tailed unpaired Student's *t*-test or by one-way ANOVA test (GraphPad) as detailed in the figure legends.

## Reporting summary

Further information on research design is available in the Nature Portfolio Reporting Summary linked to this article.

## Data availability

Source data are provided with this paper. All correspondence and requests for materials should be addressed to M.H.G.

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

## Acknowledgements

We would like to dedicate this article to our colleague Fred W. van Leeuwen who passed away during the preparation of the final version of the manuscript for publication, his visionary insight inspired us to continue the research project. We thank Rob A.I de Vos and the Laboratory for Pathology Eastern Netherlands, Hengelo, The Netherlands for procurement and Braak staging of brain samples from human donors, DY Kim for lentiviral vectors, N. Dantuma for an initial UBB[+1] construct, and P. Davies (Feinstein Institute for Medical Research, Manhasset, NY, USA) for his gift of MC1 and PHF-1 antibodies; Y. Yosefzon for discussion, advice, and technical assistance; E. Barak, S. Kirzner, R. Lerer, and N. Dahan for assistance with FACS and micro-scopy; T. Ziv and members of the THHI Proteomics facility for mass spectrometry. We aknowledge funding from ICRF acceleration and Israel Science Foundation grants (YF), Schmidt Futures and The Israel Science Foundation (755/19) (MHG), and BIRAX and Alzheimer Society (73BX16TKMG) (TK and MHG).

## Author contributions

I.M., Y.F. and M.H.G. designed the research. I.M., M.S., A.B., A.F., R.A., E.Ko., I.M.G., N.R. and D.S. carried out experiments, S.S.-Z. and D.M. provided mice tissues, T.L. and O.K. performed MS/MS sample handling, B.M.V. and F.W.v.L. provided human brain samples, A.B., I.M., N.K. and E.Ke. conducted E.M experiments, D.M. designed mice experiments, O.K. designed and analyzed MS/MS data. B.M.V. and F.W.v.L. outlined approaches with human samples. T.K. interpreted results. I.M., B.M.V., Y.F. and M.H.G. wrote the first versions of the manuscript. All authors read and commented on the manuscript.

## Competing interests

The authors declare no competing interests.
