## [Peer Review File · Nature Communications]

Altered ubiquitin signalling induces Alzheimer's disease-like hallmarks in a three-dimensional human neural cell culture modelEditorial Note: This manuscript has been previously reviewed at another journal that is not operating a transparent peer review scheme. This document only contains reviewer comments and rebuttal letters for versions considered at Nature Communications.

REVIEWER COMMENTS

Reviewer #1 (Remarks to the Author):

I'm satisfied

Reviewer #5 (Remarks to the Author):

In this review, we will focus primarily on ubiquitin related aspects. In previous review, we have indicated that the way aggregates data are presented are somewhat misleading. For example in Fig 3g, the images give the impression that there is not aggregate in UBB+1/UCH-L1, even though Fig 3h shows that there is reduced aggregates. It is more useful to show a larger field of view to give a representative image or show what a typical aggregate look like in UBB+1 vs UBB+1/UCH-L1. In addition the use of t-test is questionable for this type of assay, especially with n=3, since the assay is based on counting of events.

The authors have changed the shRNA sequence to a new sequence that could potentially distinguish between UBB and UBB+1. It is unsatisfying that they did not show that the shRNA actually knock down UBB+1 and that it does not affect UBB. The evidence presented that the shRNA can target UBB+1 is based on transfection of UBB+1. The authors pointed out that total ubiquitin did not change in their experiment. The authors fail to recognize, however, that UBB only contribute a small fraction of the total ubiquitin in neurons and probably less in other cells under normal conditions. UBB is a stress inducible gene and its expression might increase under stress conditions. In Fig 4c, the levels of UBB+1 and ubiquitin should be shown.

Along the same line the authors do not describe the antigen used to generate the UBB+1 antibody. This information is crucial to provide to the reader. Assuming it is raised against the C-terminal extension, the authors still need to show that it is specific to UBB+1. In addition, the cropping of images to show only one band for UBB+1 is not helpful as UBB+1 clearly exists with different molecular weights based on Fig 3c and 3d, for example.

As pointed in earlier review, the data for UCH-L1 ubiquitinating APP in Fig 3f are very weak. The data with UCH-L1 and UBB+1 combination are very confusing. For example, UBB+1 supposedly increase APP by inhibiting UCH-L1. However, treating cells with LDN57444 did not result in the same change in APP as UBB+1. If it is an issue of extent, it is not clear why LDN57444 and UBB+1 together also fail to cause the same change.

Reviewer #1 (Remarks to the Author):

I'm satisfied

Reviewer #5 (Remarks to the Author):

In this review, we will focus primarily on ubiquitin related aspects. In previous review, we have indicated that the way aggregates data are presented are somewhat misleading. For example, in Fig 3g, the images give the impression that there is not aggregate in UBB+1/UCH-L1, even though Fig 3h shows that there is reduced aggregates. It is more useful to show a larger field of view to give a representative image or show what a typical aggregate look like in UBB+1 vs UBB+1/UCH-L1. In addition the use of t-test is questionable for this type of assay, especially with n=3, since the assay is based on counting of events.

In Figure 3g, just like in other representative images of aggregates in 3D neuronal culture (such as 2b, 4a and the original papers: Choi, S.H. et al. Nature 515, 274-278 (2014), Or Kim, Y.H. et al Nat Protoc 10, 985-1006 (2015)) the purpose of the image was to show an example of what was counted. It was impractical to show a larger field because the experiments were performed in a 24-well plate and the aggregates were sparse and widely spread out; at low magnification, the resolution is not informative. Regretfully, we did not take a picture of a specific aggregate from the UBB+1/UCHL1 culture, though we do have additional ones for the UBB+1 culture. We wish to emphasize that the quantification of the aggregates in Figure 3h was performed in a double-blind manner (two scientists counting fields without knowing which sample they were given by the other) on an entire well (24 well plates); in the case of UBB+1/UCHL1 this amounted to ~10 aggregates on average.

In addition, following the reviewer's suggestion, we performed a one-way ANOVA test for this experiment. The manuscript is updated accordingly.

The authors have changed the shRNA sequence to a new sequence that could potentially distinguish between UBB and UBB+1. It is unsatisfying that they did not show that the shRNA actually knock down UBB+1 and that it does not affect UBB.

We have shown in extended data that the shRNA knocks-down UBB+1 (ext. data fig. 5a-in HEK293 cells and ext. data fig. 5d, e- in the 3D system). Initially, we confirmed that this shRNA does not affect total ubiquitin in HEK293 cells (ext. data 5a,b). For this revision, we also show that shRNA does not affect total ubiquitin in the 3D neuronal cultures (new extended data Figure 5F) and go further by extracting RNA from 3D FAD cultures and provide new evidence by qPCR that UBB mRNA levels are unchanged by shUBB+1 (new ext. fig.5g). We wish to mention that there are published examples of siRNAs that can differentiate between genes base don a single nucleotide doi.org/10.1371/journal.pgen.0020140.

The evidence presented that the shRNA can target UBB+1 is based on transfection of UBB+1. The authors pointed out that total ubiquitin did not change in their experiment. The authors fail to recognize, however, that UBB only contribute a small fraction of the total ubiquitin in neurons and probably less in other cells under normal conditions. UBB is a stress inducible gene and its expression might increase under stress conditions. In Fig 4c, the levels of UBB+1 and ubiquitin should be shown.

To answer the reviewer, we extracted proteins from eight-week-old 3D FAD SCR vs. FAD shUBB+1 cultures, resolved proteins by SDS PAGE, and performed an immunoblot assay for ubiquitin. We include the new information that total ubiquitin levels are not affected by shUBB+1 (new ext. fig.5f). In addition, we show that total ubiquitin protein levels are not changed due to the putative stress that expression of FAD proteins or of UBB+1 may induce (new ext. fig. 3f).

Along the same line the authors do not describe the antigen used to generate the UBB+1 antibody. This information is crucial to provide to the reader. Assuming it is raised against the C-terminal extension, the authors still need to show that it is specific to UBB+1. In addition, the cropping of images to show only one band for UBB+1 is not helpful as UBB+1 clearly exists with different molecular weights based on Fig 3c and 3d, for example.

The sequence of the antibody raised against UBB+1 was taken from D.F. Fischer, et al., , FASEB J, 17 (2003) 2014-2024. We note that the antibody is against the entire missense extension

YADLREDPDRQDHHPGSGAQ that shares no sequence overlap with the ubiquitin primary sequence. Moreover, the specificity for UBB+1 (in contrast to ubiquitin) has been confirmed by multiple techniques (D.F. Fischer, et al. FASEB J, 17 (2003) 2014-2024; F.W. van Leeuwen, et al. Science, 279 (1998) 242-247).

In addition, in light of the reviewer's comment, we included the uncropped version of Extended Data Fig, 5a displaying the specificity of the anti-UBB⁺¹ antibody (right panel), the slightly higher MW modified form of UBB⁺¹ (right panel), and the dramatic potency of shUBB⁺¹ specifically on UBB⁺¹ in contrast to all other ubiquitin specific and conjugates in whole cell extract (left panel).

As pointed in earlier review, the data for UCH-L1 ubiquitinating APP in Fig 3f are very weak. The data with UCH-L1 and UBB+1 combination are very confusing. For example, UBB+1 supposedly increase APP by inhibiting UCH-L1. However, treating cells with LDN57444 did not result in the same change in APP as UBB+1. If it is an issue of extent, it is not clear why LDN57444 and UBB+1 together also fail to cause the same change.

The purpose of this experiment was to show that UBB⁺¹ causes an elevation in APP protein levels while decreasing its ubiquitination. The juxtaposition with UCHL1 is fascinating since UCHL1 depletes APP levels while increasing its ubiquitination. Initially, we tried to compare the effects of UBB+1 with that of a specific chemical inhibitor of UCHL1, LDN57444, in order to imply that the effect of UBB+1 on APP may be via UCHL1. We toned down the attention to this suggested mechanism in the text.

REVIEWERS' COMMENTS

Reviewer #5 (Remarks to the Author):

The authors have addressed my concerns.